# Application of Artificial Intelligence in Orthodontics: Current State and Future Perspectives

**DOI:** 10.3390/healthcare11202760

**Published:** 2023-10-18

**Authors:** Junqi Liu, Chengfei Zhang, Zhiyi Shan

**Affiliations:** 1Division of Paediatric Dentistry and Orthodontics, Faculty of Dentistry, The University of Hong Kong, Hong Kong SAR, China; u3009390@connect.hku.hk; 2Division of Restorative Dental Sciences, Faculty of Dentistry, The University of Hong Kong, Hong Kong SAR, China; zhangcf@hku.hk

**Keywords:** artificial intelligence, orthodontics, machine learning, deep learning

## Abstract

In recent years, there has been the notable emergency of artificial intelligence (AI) as a transformative force in multiple domains, including orthodontics. This review aims to provide a comprehensive overview of the present state of AI applications in orthodontics, which can be categorized into the following domains: (1) diagnosis, including cephalometric analysis, dental analysis, facial analysis, skeletal-maturation-stage determination and upper-airway obstruction assessment; (2) treatment planning, including decision making for extractions and orthognathic surgery, and treatment outcome prediction; and (3) clinical practice, including practice guidance, remote care, and clinical documentation. We have witnessed a broadening of the application of AI in orthodontics, accompanied by advancements in its performance. Additionally, this review outlines the existing limitations within the field and offers future perspectives.

## 1. Introduction

AI is a subfield of computer science that refers to the ability of a machine to imitate cognitive functions of human intelligence [1]. Over the last decade, the field of AI has shown a lot of potential as it can be employed to solve a variety of tasks. The expert system and machine learning are two important branches of AI. Unlike the knowledge-based expert system, which is established based on predetermined rules and knowledge, machine learning focuses on “learning” from training data to improve its capability [2,3]. In addition to its strong adaptability and generalization capabilities, machine learning is capable of processing large-scale data and has more open-source algorithms, which makes it one of the most promising technologies in AI.

Artificial neural networks (ANNs), a sub-domain of machine learning, draw inspiration from the biological neural system of the human brain [4]. ANNs have been notably employed to analyze intricate connections between massive data [5]. An ANN typically has a minimum of three layers, namely, an input layer, an output layer, and at least one hidden layer [6]. Neurons within each layer are interconnected to establish a network of processors. ANNs encompassing multiple hidden layers are commonly referred to as deep learning, which has demonstrated exceptional performance in computer vision tasks such as classification and segmentation [7]. Deep learning is becoming increasingly popular due to its high feasibility and growing computing performance, as well as advanced model training algorithms [8]. In addition, one notable advantage of deep learning over traditional machine learning is that it allows automated feature extraction without manual intervention, enabling the better harnessing of the information within the data [9]. Convolutional neural networks (CNNs), one of the most widely used deep learning algorithms, exhibit particularly remarkable performance in handling high-resolution images [10,11,12]. In CNN, the hidden layers are substituted with three distinct functional layers, namely, convolutional layers, pooling layers, and fully connected layers. The convolutional layers employ convolutional kernels as filters to generate feature maps. The convolution process effectively reduces image complexity, making CNNs highly suitable for tasks like recognizing objects, shapes, and patterns. The pooling layers are commonly employed after convolutional layers to decrease the dimension of feature maps while retaining essential information. Following several iterations of convolutional and pooling layers, the outputs are integrated in the fully connected layers for further decision making. Consequently, thanks to the abovementioned three layers, CNNs outperform algorithms such as ANNs in image-related tasks [6,11].

Malocclusion is distinguished by an anomaly in teeth alignment, occlusion and/or craniofacial relationships [13]. It is a deviation from the norm and the manifestation of normal biological variability [14]. Numerous studies have indicated that the presence of malocclusion not only affects oral health and dental aesthetics but also has a negative impact on psychological well-being and social interactions [15,16,17]. Malocclusion is considered the world’s third most prevalent oral disease, and nearly 30% of the population present with great need of orthodontic treatment [18,19]. Clinical orthodontic practice often requires a significant amount of time to conduct various analyses that necessitate the extensive clinical experience of orthodontists. These workloads have affected the efficiency of clinical orthodontic practice and have also made orthodontic treatment less accessible for non-specialists due to the requirement for clinical experience.

A series of studies have shown that AI can significantly enhance the efficiency of clinical orthodontic practice [20,21]. Several commercially available AI-driven software (3Shape Dental System 2.22.0.0, Uceph 4.2.1, Mastro 3D V6.0 etc.) programs have found widespread applications in orthodontic care. With the ongoing advancement of AI algorithms, computing capabilities and the growing availability datasets, the scope of AI applications in orthodontics is expanding, accompanied by continuous performance improvement. Staying updated on the latest developments of AI applications in orthodontics through timely summaries helps researchers gain a rapid and accurate understanding of this field. In addition, despite obtaining encouraging results, there is still significant room for progress in the application of AI in orthodontics. Therefore, this review provides a comprehensive summary of the current state of AI applications in orthodontics, encompassing diagnosis, treatment planning, and clinical practice. Additionally, the review discusses the current limitations of AI and offers future perspectives, aiming to offer valuable insights for the integration of AI into orthodontic practice.

## 2. Application of AI in Orthodontics

### 2.1. Diagnosis

A satisfactory orthodontic diagnosis relies on a series of analysis, like cephalometric analysis, dental analysis, facial analysis, skeletal maturation determination and upper-airway obstruction assessment, to comprehensively evaluate patients’ overall profile, including their facial profile, dental and skeletal relationship, skeletal maturation stages and upper-airway patency [22].

#### 2.1.1. Cephalometric Analysis

Cephalometric analysis, especially landmarking on lateral cephalograms, serves as the foundation of orthodontic diagnosis, treatment planning and treatment outcome assessment. Conventional manual landmarking is time-consuming, experience-dependent and can be inconsistent within and across orthodontists, significantly affecting the efficiency and accuracy of clinical practice [23,24,25,26]. The automated landmark detection was reported as early as the mid-1980s, but the error margin was too high to be implemented in clinical practice [27]. In recent years, with the advancement of AI, numerous studies have been conducted using cephalometric analysis, the reproducibility, efficiency, and accuracy of which are continuously being enhanced [24,28,29,30,31,32,33,34,35,36,37,38,39,40,41,42,43,44,45,46,47,48,49,50,51,52,53,54,55,56,57,58,59,60,61,62,63,64,65]. Notably, cephalometric analysis has emerged as the most extensively explored domain of AI applications in orthodontics. Given the vast amount of the relevant literature available, it is impractical to list all of the relevant literature about automated cephalometric analysis. Consequently, this review only summarizes the pertinent literature published within the past five years, as depicted in Table 1, in order to offer the latest advancements in this field [24,28,29,30,31,32,33,34,35,36,37,38,39,40,41,42,43,44,45,46,47,48,49,50,51,52,53,54,55,56,57,58,59,60,61,62,63,64,65].

In general, acceptable linear and angular measurement are less than 2 mm and 2°, respectively [23,36,38,43,44,47,54,66,67,68,69,70]. Following this criterion, although some commercially available software can achieve high overall accuracy in automated landmarking on lateral cephalograms, manual supervision is still recommended [47,48,53,54,56].

Compared to classical machine learning, deep learning, especially CNNs, demonstrates superior performance and has been investigated more (Table 1). Several studies have shown that You-Only-Look-Once version 3 (YOLOv3), a popular CNN algorithm, has yielded remarkable results in automated landmarking [32,33,43,44]. Park et al. compared the accuracy and computational efficiency of two CNN algorithms, YOLOv3 and the Single-Shot Multibox Detector (SSD), in identifying 80 landmarks in lateral cephalometric radiograph images. The results indicated that YOLOv3 exhibited superior accuracy and computational performance compared to SSD [31]. To mitigate the risk of overfitting and enhance the generalizability of data, Kim et al. collected 3150 lateral cephalograms taken by nine different cephalography machines from multiple centers nationwide. The researchers utilized the cascade CNN algorithm and achieved an overall automated detection error of 1.36 ± 0.98 mm [24]. The same team developed a CNN algorithm that reached 1.17 mm of total mean error in lateral cephalogram landmark identification despite the presence of genioplasty, bone remodeling, and orthodontic and orthognathic appliance, paving the way for its further use in orthognathic surgical patients [46]. Yao et al. utilized a CNN-based model to identify 37 landmarks in lateral cephalograms, reaching 1.038 ± 0.893 mm of MRE and 97.30% of SDR within 2 mm [50]. To the best of our knowledge, this model achieved the best performance in automated landmarking. The existing CNN models do have some drawbacks, such as down-sampling quantization errors, and requiring preprocessing or postprocessing to improve accuracy, which may increase computational cost and time. To address these issues, Lu et al. proposed three-layer graph convolutional networks (GCNs), obtaining 1.19 mm of MRE and 83.20% of SDR within 2 mm [51].

At the same time, research has reported the use of CNNs for automated landmarking on posteroanterior cephalograms to assess mandibular deviation, which can aid in evaluating facial symmetry [57,58]. Thanks to the advancement of computational power, AI has also made progress in three-dimensional (3D) cephalometric landmark detection, and deep learning and CNNs are the most efficient methods [59,60,71,72,73]. Blum et al. utilized a CNN-based model to conduct 3D cephalometric analysis, which yielded a mean error of 2.73 mm and exhibited a 95% reduction in processing time compared with manual annotation [65]. Dot et al. proposed a fully CNN, SpatialConfiguration-Net, for the 3D automated detection of 33 landmarks and 15 measurements, achieving superior outcomes. Specifically, the MRE for landmarks was only 1.0 ± 1.3 mm, and SDR within 2 mm reached 90.4%. Regarding its measurements, the mean errors were −0.3 ± 1.3°and −0.1 ± 0.7 mm for its angular and linear variables [64]. Deep reinforcement learning (DRL), the algorithm that merges the advantages of deep learning (perception ability) and reinforcement learning (decision-making ability), has also garnered attention for its performance in 3D localization [74,75,76]. Kang et al. utilized multiple-stage DRL for 3D automated landmark detection. The DRL algorithm achieved a mean detection error of 1.96 ± 0.78 mm for landmarks, with 58.99% and 95.70% of the detection rate falling within the 2 mm and 4 mm range [62]. Nevertheless, the current progress in automated 3D cephalometric analysis is predominantly concentrated on landmark detection, with limited emphasis on linear and angular measurements. It is anticipated that future advancements will address this limitation.

#### 2.1.2. Dental Analysis

In orthodontic clinical practice, the utilization of intraoral photographs and orthodontic study models is imperative for dental analysis. These examinations provide clinicians with comprehensive information regarding various aspects, including molar relationships, tooth crowding, dental arch width, overjet and overbite, and oral health status. However, the manual analysis of these examinations is both time-consuming and labor-intensive. Consequently, there is potential for AI to replace human involvement in this analysis. Talaat et al. utilized the YOLO algorithm to detect malocclusion (specifically tooth crowding or spacing, abnormal overjet or overbite, and crossbite) from intraoral photographs. The results showed an exceptional accuracy rate of 99.99% [77]. Similarly, using intraoral imaging as training data, Ryu et al. utilized four CNN algorithms to assess tooth crowding; the results showed that VGG19 had the minimum mean errors in the maxilla (0.84 mm) and mandible (1.06 mm) [78].

The development of digital technology has significantly facilitated the adoption of 3D intraoral scanner images and digital dental models in clinical practice. Some companies, such as Invisalign (Align Technology, Santa Clara, CA, USA), have effectively utilized 3D oral scan data and digital models for automated measurement and analysis. In addition, Im et al. proposed a dynamic-graph convolutional neural network (DGCNN) to automate tooth segmentation in digital models, achieving superior accuracy and reduced computational time compared to the other two commercially available pieces of software: OrthoAnalyzer (ver.1.7.1.3) and Autolign (ver.1.6.2.1) [79]. Beyond that, the accurate segmentation of teeth and the recognition of landmarks on teeth are crucial for automated dental analysis, and significant advancements have been consistently achieved in this domain, hopefully paving the way for further clinical applications [80,81,82,83,84].

#### 2.1.3. Facial Analysis

Facial photographs play a pivotal role in evaluating facial asymmetry and proportions. To our knowledge, so far, there are only three articles which have reported on automated facial analysis, and all of them used 2D frontal photos as training data.

Rao et al. utilized an active shape model algorithm for automated landmarking and measuring on facial images, but only just over 50% of the landmark measures had an error within 3 mm [85]. Yurdakurban et al. compared a machine-learning-based software with researchers in detecting facial midline and evaluating asymmetry, and the differences in most measurements between the two methods were not statistically significant [86]. Rousseau et al. employed a CNN to analyze the vertical dimension of patients. The results showed higher precision and efficiency than manual measurements, with the 95% confidence interval limit of agreements between the manual and automated methods less than 10% [87]. Overall, automated facial analysis is still in its early stages and requires further research to improve its accuracy and applications.

#### 2.1.4. Skeletal Maturation Determination

The determination of patients’ growth spurt is critical for orthodontic treatment, especially for those that need functional and orthopedics treatment. Hand–wrist X-rays have been regarded as the most conventional and accurate way to determine skeletal age. In recent years, several studies have reported combining AI with hand–wrist radiographs to predict skeletal age [88,89,90]. A number of research studies have revealed that the cervical vertebral maturation (CVM) method is also effective for growth estimation and highly correlates with the hand–wrist radiograph method [91,92,93,94,95,96]. Therefore, to minimize unnecessary radiation exposure, hand–wrist X-ray is not routinely used in clinical orthodontic practice [97,98]. Instead, the CVM method, which evaluates the size and shape of the cervical vertebrae through lateral cephalograms, has become increasingly popular in predicting skeletal maturation [91,92,93,94,95,96]. The application of AI in skeletal maturation assessment using lateral cephalograms was summarized in Table 2 [91,98,99,100,101,102,103,104,105,106,107].

Kök et al. utilized seven different machine-learning algorithms to determine the CVM stages [99]. The results showed that these algorithms exhibited varying levels of accuracy in predicting different CVM stages, but the ANN was considered the most stable algorithm, with an average rank of 2.17 in determining all the CVM stages [99]. Similarly, Amasya et al. developed and compared the performance of five ML algorithm in CVM analysis, and the ANN model proved to be better at classification than the other four algorithms (decision tree, random forest, logistic regression, and support vector machine) [101]. The same team further compared this ANN model with four independent human observers to automate the cervical vertebral maturation stages but only reached an average of 58.3% agreement with the observers [102]. Several studies have employed CNN models for CVM prediction, and different degrees of accuracy were obtained [98,105,106]. Makaremi et al. pointed out that an equal distribution of images across all CVM stages is beneficial for improving CNN accuracy [100]. Zhou et al. increased the sample size to enhance the reliability of the results [104]. Seo et al. pioneeringly compared six unsupervised CNN models and utilized a gradient-weighted class activation map (Grad-CAM) to visualize the models [103]. The results indicated that all the algorithms achieved an accurate rate of over 90%, with Inception-ResNet-v2 showing the best performance at 0.941 ± 0.018% accuracy. In addition, the Grad-CAMs showed that Inception-ResNet-v2 focused on several cervical vertebrae, unlike most of the other algorithms that mainly focused on the third cervical vertebra. Radwan et al. also used both the CNN model and unsupervised learning method to predict CVM stages, with a larger sample size and a validation dataset to tune the algorithm. However, the classification network only obtained an accuracy of 0.802 [107]. In summary, ANNs have received much attention and recognition in the early years, but in recent years, CNNs have gradually become more prominent in image-related tasks. With continuous improvements in algorithms, it is expected to achieve more encouraging results in the future.

#### 2.1.5. Upper-Airway Obstruction Assessment

Skeletal deformity and airway obstruction mutually influence each other. Upper-airway obstruction can alter breathing, which can affect the normal development of craniofacial structures and potentially lead to malocclusion and other craniofacial abnormalities. [108]. Screening the presence of upper-airway obstruction, especially adenoid hypertrophy, is critical for orthodontic diagnosis and treatment planning. The application of AI in upper-airway obstruction assessment is summarized in Table 3 [109,110,111,112,113,114,115,116,117].

Detecting adenoid hypertrophy based on lateral cephalograms has been proven to be highly accurate and reliable [118,119]. The adenoid–nasopharyngeal (AN) ratio proposed by Fujioka is the most notable method [120]. Both Shen et al. and Zhao et al. employed a CNN model to locate four key points in Fujioka’s method on lateral cephalograms, and subsequently calculated the AN ratio [109,110]. The proposed model by Shen et al. obtained a classification of 95.6% and a mean AN ratio error of 0.026 [68]. The model of Zhao et al. also showed favorable performance, with high accuracy (0.919), sensitivity (0.906) and specificity (0.938) [98]. Liu et al. utilized VGG-Lite to directly detect adenoid hypertrophy on lateral cephalograms without automated landmarking, and the model achieved a positive predictive value of 0.880 and negative predictive value of 0.900 [111]. Dong et al. proposed two deep learning algorithms, the hierarchical masks self-attention U-net (HMSAU-Net) and 3D-ResNet, to automatically segment upper airways and diagnose adenoid hypertrophy, respectively, from CBCT. Of note, a high accuracy of 0.912 was achieved by the adenoid hypertrophy diagnosis model [116].

In addition to adenoid hypertrophy, the morphology and volume of the upper airway are also important indicators for assessing upper-airway obstruction. By using a CNN model, Jeong et al. obtained promising results in automated upper-airway obstruction evaluation based on lateral cephalograms, with a positive predictive value of 0.90 and negative predictive value of 0.85 [115]. The segmentation of the airway from CBCT can provide a 3D view, enabling the more accurate detection of airway obstruction. Recent studies have shown continuous progress in airway segmentation, with deep learning, especially CNN algorithms, being the most commonly used. Sin et al. generated a CNN algorithm to automatically segment and calculate the volume of a pharyngeal airway from CBCT images, achieving a dice ratio of 0.919 and a weighted Intersection over Union (IoU) of 0.993 [112]. Shujaat et al. employed the 3D U-Net and obtained an accuracy of 100% in segmenting a pharyngeal airway [114]. Jin et al. utilized a transformer and U-Net-based model and segmented the entire upper airway, including the nasal cavity and hypopharynx [117].

### 2.2. Treatment Planning

Orthodontic treatment requires cautious decision-making processes that are the cornerstone of a satisfactory treatment outcome, such as tooth extraction plan and the possibility of surgical interventions. AI is expected to assist orthodontists especially those inexperienced in making correct decisions. The application of AI in treatment planning is summarized in Table 4 [78,121,122,123,124,125,126,127,128,129,130,131,132,133,134,135,136,137,138,139,140,141].

#### 2.2.1. Decision Making for Extractions

Currently, there is no absolute standardized formula for extraction diagnosis and patterns, and the decision depends, to some extent, on the orthodontists’ experience [142]. A wrong decision about extraction could cause a series of irreversible problems like an unfavorable profile, improper occlusion and extraction-space closure difficulties. AI can contribute to reducing the likelihood of incorrect tooth extraction protocols.

ANNs are the most utilized method to predict extraction diagnosis and patterns [121,122,123,124,125,126]. Jung et al. built an AI expert system with neural-network machine learning based on 12 cephalometric variables and 6 additional indices, reaching a success rate of 93% and 84% in deciding extraction/non-extraction and detailed extraction patterns, respectively. In this study, one-third of the learning dataset was chosen as the validation set to prevent overfitting [122]. Li et al. adopted a multilayer perceptron ANN and obtained similar results, with an accuracy of 94% and 84.2% in the determination of extraction diagnosis and patterns. In addition, the proposed algorithm achieved an accuracy of 92% in predicting anchorage patterns [123].

Different machine-learning algorithms have their own strengths and weaknesses. For example, random forest (RF) and support vector machine (SVM) are often used for classification and regression tasks, and logistic regression (LR) is more suitable for binary classification tasks [143,144,145]. Several studies have used different machine-learning algorithms to determine tooth extraction plans [124,125,126,128]. The research results of Shojaei et al. indicated that compared to some traditional machine-learning algorithms, ANNs demonstrated significant advantages in decision making for extraction and anchorage patterns [126]. Leavitt et al. compared three machine-learning algorithms (RF, LR, and SVM) for predicting extraction patterns, but their overall accuracies were not very satisfactory, with SVM achieving the highest accuracy at 54.55% [128]. Although RF can act as an ensemble method to prevent overfitting and performed well in some studies, more research is still needed to further substantiate its effectiveness [124,125]. The abovementioned models used manual measurement values instead of images as the input data. Recently, Ryu et al. employed intraoral photographs and extraction decisions as the input data and utilized four CNN algorithms to build a tooth extraction prediction model. The results indicated VGG19 had the highest prediction accuracy in both the maxilla (0.922) and mandible (0.898) [78]. In summary, there have been several studies which have used AI for decision making during extractions. Most of these studies have used the extracted measurements of patients as their input data. However, the varying number of measurements used in different studies may result in relatively low comparability between the results. Overall, ANNs have shown the best performance in decision making for extractions. However, with changes in the input data type, such as radiographic images, other algorithms like CNNs may exhibit better performances.

#### 2.2.2. Decision Making for Orthognathic Surgery

For adult patients with severe dentofacial deformities, combined orthodontic and orthognathic surgical treatment is usually required to reposition the jaws. Currently, there is no absolute criterion for determining surgical cases, especially in borderline situations where the dilemma between camouflage orthodontic treatment and surgical treatment often confuses inexperienced orthodontists [146,147,148].

Lateral cephalograms are the most used method in clinical practice to assess sagittal skeletal deformities. Several studies have used lateral cephalograms as the input data, whether using an ANN or CNN, and all achieved accuracy rates exceeding 90% [131,132,135]. Shin et al. adopted both lateral cephalograms and posteroanterior cephalograms as their training data to take both the sagittal and lateral relationship of the jaws into consideration [134]. The proposed CNN model reached an accuracy of 95.4% in predicting orthognathic surgery diagnosis.

Facial appearance is also a crucial factor when making the surgical/non-surgical decision. Knoops et al. utilized SVM to predict a surgery/non-surgery decision based on 3D facial images, and reached an accuracy of 95.4% [130]. Trained by front and right facial photos, the CNN model proposed by Jeong et al. only showed an accuracy of 89.3% [133]. Choi utilized a variety of factors as the training data, including lateral cephalometric variables, dental variables, profile variables, and the chief complaint for protrusion [131]. The proposed ANN model not only predicted the surgery/non-surgery decision but also anticipated in the tooth extraction plan for the surgical cases, obtaining an accuracy ranging from 88% to 97%. Nevertheless, it is worth noting that this study did not encompass Class I surgical cases, which may have influenced the generalizability of the model [131]. Using similar types of input data, Lee et al. investigated the abilities of RF and LR to predict the surgery decision of Class III patients, but they only obtained an accuracy of 90% (RF) and 78% (LR) [136]. Overall, AI has made some progress in decision making for orthognathic surgery. However, there is still a need for further improvement in incorporating a more comprehensive type of cases, especially more-borderline cases, which holds the promise of enhancing AI’s diagnostic capabilities.

#### 2.2.3. Treatment Outcome Prediction

For some cases, orthodontists may develop more than one treatment plan, especially for borderline cases. However, deciding on the most suitable treatment plan can be challenging for inexperienced orthodontists. In addition, the treatment outcome for cases involving extraction and interproximal enamel reduction is often irreversible, and suboptimal plans may result in patient dissatisfaction. Predicting treatment outcomes can help orthodontists analyze and treat malocclusions more scientifically, reducing potential risks and complications during and after clinical treatment. Currently, AI can aid in predicting dental, skeletal and facial changes, as well as patients’ experience of clear aligners, thereby guiding the treatment planning [137,138,139,140,141].

Orthodontic tooth setup, initially proposed by Kesling, enables the visualization of the treatment progress and final occlusion, but manual tasks like tooth segmentation and reposition are labor-intensive. With the continuous advancements of digital orthodontics and artificial intelligence, automated virtual setups have been widely applied, especially in the field of clear aligners [137]. Woo et al. compared the accuracy of three pieces of automated digital-setup software with that of a manual setup regarding six directions of tooth movement [137]. The results indicated that the pieces of automated virtual-setup software were effective overall, but further manual adjustments may be still required in clinical practice. Also, it is important to note that the study only included cases where no extractions were performed.

In addition to dental changes, there have been several studies using AI to predict skeletal and facial changes after orthodontic treatment. Park et al. applied a CNN model to predict the cephalometric changes of Class II patients after using modified C-palatal plates, and obtained an overall accuracy of 1.79 ± 1.77 mm [138]. Tanikawa et al. combined geometric morphometric methods and deep learning to predict 3D facial-morphology change after orthodontic (with four premolars extracted) or orthognathic surgical treatment [139]. The proposed system showed an average error of 0.94 ± 0.43 mm and 0.69 ± 0.28 mm in the surgery and orthodontic group, respectively. In another study, a conditional generative adversarial network (cGAN), a type of deep learning algorithm, was used to predict 3D facial changes after orthodontic treatment based on patients’ gender, age and incisor movement [140]. Thanks to the conditions applied to the generator and discriminator, cGAN is supposed to generate high-quality image samples and excels at performing image-to-image translational tasks [149,150]. As a result, 3D facial images and color distance maps were generated, and the distances of six perioral landmarks between the real model and predicted model were calculated, with the cGAN achieving a mean prediction error of 1.2 ± 1.01 mm and an accuracy (within 2 mm) of 80.8% [140].

The selection of the treatment appliance is a crucial aspect of orthodontic treatment planning. Particularly for patients using clear aligners, a poor wearing experience can impact patients’ compliance and consequently affect the treatment outcomes. Xu et al. utilized 17 clinical features as the training data and employed an ANN model to predict patients’ experiences of Invisalign treatment [141]. The proposed model achieved high prediction accuracies of 87.7% for pain, 93.4% for anxiety, and 92.4% for quality of life. To the best of our knowledge, this was the first and currently the only study that utilized AI to predict patient experience of orthodontic treatment, laying the foundation for further research in this area. However, a limitation of this study is that it only included patients’ clinical features as input data and did not incorporate other potential influencing factors such as gender and education level, which could potentially affect the predictive ability of the model [151].

### 2.3. Clinical Practice

During the orthodontic treatment, orthodontists often come across various challenges, including clinical expertise in orthodontics and patient communication and management. The application of AI can help facilitate efficient and effective orthodontic treatment regarding practice guidance, remote care and clinical documentation [152,153,154,155,156,157,158,159,160].

#### 2.3.1. Practice Guidance

A deep overbite is one of the most common and challenging malocclusions to correct in orthodontic treatment [161]. El-Dawlatly et al. proposed a computer-based decision support system for deep-overbite treatment guidance, trained by the actual treatment changes [152]. Instead of answering binary questions, the model can provide a detailed treatment protocol on deep-overbite correction from seven aspects, such as the intrusion or proclination of incisors, leveling the curve of Spee. With a high success rate of 94.40%, this model is expected to aid orthodontics in correcting deep overbites in the future.

The 3D U-Net, a deep learning algorithm, is widely used in 3D image segmentation. As a modified version of 3D U-Net, 3D U-Net with squeeze-and-excitation modules (3D-UnetSE) has achieved better performance in capturing high level features [153,162]. The stability of palatal mini implants is associated with hard and soft tissues [163,164]. Tao et al. successfully used 3D-UnetSE to accomplish the automated segmentation and thickness measurement of palate bone and soft tissue through CBCT. Furthermore, ideal sites for palatal miniscrews were predicted based on the bone and soft tissue thickness [153].

Monitoring the tooth root position throughout the orthodontic treatment is essential to better prevent adverse outcomes and assess treatment effectiveness. However, conventional methods, whether CBCT or panoramic films, increase radiation exposure. Hu et al. and Lee et al. used deep learning to accurately segment teeth in CBCT scans and merged the segmented teeth with intraoral scanned dental crowns to construct integrated tooth models [154,155]. In this way, orthodontists can determine the position of tooth roots solely based on intraoral scans. These two studies showed the promising performance of tooth position prediction; with continuous improvement in the accuracy of tooth segmentation, integrated models are expected to be widely applied in clinical practice.

#### 2.3.2. Remote Care

Remote monitoring allows orthodontists to remotely track treatment progress and provide timely feedback based on photos or oral scans of the dentition, avoiding unnecessary visits, and bringing flexibility and convenience to patients [156,157,158].

AI has enhanced the applications and effectiveness of remote monitoring software [156]. Dental monitoring (DM, Paris, France), standing out as one of the leading software in AI-driven remote monitoring, has gained widespread popularity and research attention [157,165]. DM is user-friendly, allowing patients to scan their dentitions using a smartphone. Studies have indicated that DM can not only reduce chairside time, but also improve patients’ compliance [156,157]. DM can be applied to both conventional fixed appliances and clear aligners, automatically detecting numerous metrics, such as ill-fitting clear aligners, losses of attachments, archwire passivity, bracket breakages and relapse occurring [158,166,167]. In addition, DM’s detections demonstrate a high level of precision. Homsi et al. claimed that the remotely reconstructed digital models generated by DM were as highly accurate as intraoral scans [168] Moylan et al. reached similar viewpoints by measuring intercanine and intermolar width DM-generated models and plaster models [169]. However, a recent prospective study found that there are still problems with the consistency of DM instructions, especially for the determination of teeth with tracking issues. At the same time, the rationale for DM instructions for clear aligner replacement is difficult to explain [170]. Therefore, orthodontists may adopt a cautious approach towards the widespread use of AI-driven remote monitoring tools.

#### 2.3.3. Clinical Documentation

Clinical photos and radiographs are routinely taken for diagnosis and treatment monitoring. AI can aid in classifying and categorizing these images, thereby enhancing the efficiency of clinical practice. Ryu et al. applied CNNs to automatically classify facial and intraoral photographs, including four facial photos and five intraoral photos. The CNN model obtained an overall valid prediction rate of 98% [159]. Li et al. employed a Deep hidden IDentity (DeepID)-based deep learning model and expanded the categories of orthodontic images into a total of 14, comprising 6 facial images, 6 intraoral images, 1 panoramic film and 1 lateral cephalogram. The proposed model used deep convolutional networks for feature extraction and joint Bayesian for the verification process. As a result, the DeepID model not only reached a high accuracy of 99.4% but also significantly improved the computational speed [160].

## 3. Limitations and Future Perspectives

The continuous evolution of AI has brought significant advancements in its application in orthodontics. In this review, we comprehensively introduced the recent advances in the application of AI in orthodontics, including diagnosis, treatment planning, and clinical practice. These studies suggest that the application of AI in orthodontics has made promising progress and has great potential for wider clinical applications in the near-future. However, there are still some limitations that may preclude the envisioned application of AI in orthodontics.

Firstly, the scarcity and low generalizability of training data render the current research less reliable. Taking studies incorporated in this review as an example, some AI models used to assist decision making did not include a diverse range of representative case types in the training data; although obtaining promising accuracy, their prediction for those rare deformity types is questionable. Obtaining a significant amount of high-quality data remains challenging, especially data that require manual annotation by experienced experts. A series of measures are expected to alleviate the severity of data insufficiency, such as transfer learning, data augmentation, semi-supervised learning and few-shot learning. However, the effectiveness of these methods remains limited [171,172]. Transfer learning refers to applying pre-trained models in a different but related domain, thereby reducing the dependence on extensive training data. However, this approach may exhibit low generalization capabilities when applied to a new domain [173]. Data augmentation can increase sample size through altering characteristics of existing data or generating synthetic images, but it cannot improve the diversity in biologic variability [171,174]. Semi-supervised learning is suitable when annotated data are limited, but high quality of the annotated data and enough unannotated data are still required [171]. For few-shot learning, its lack of specialized data and standard evaluation frameworks may hinder its further application [172]. Nowadays, due to ethical concerns and data protection issues, data sharing is still challenging. AI models trained with data of low generalizability would be biased. Federated learning is a distributed and decentralized machine-learning approach that allows cross-site collaboration without sharing data directly [175,176]. Blockchain, as a transparent, secure and immutable distributed shared database, provides a secure platform for data sharing and storage [177,178]. The combination of blockchain technology and federated learning is expected to facilitate data sharing through multisite collaboration without compromising data privacy, thereby providing large and more-diverse datasets [179,180].

Secondly, while a considerable amount of the literature has explored the application of AI in orthodontics, it remains challenging to directly compare different studies due to variations in study designs, dataset sizes, and evaluation metrics. To address this issue, Norgeot et al. proposed minimum-information-about-clinical-artificial-intelligence modeling (MI-CLAIM) in order to introduce comparable degrees of transparency and effectiveness to clinical AI modeling [181]. The MI-CLAIM checklist not only facilitates the assessment of the clinical impact of AI study but also enables researchers to replicate the technical design process rapidly.

In addition, despite the impressive performance of AI algorithms, particularly deep learning, their lack of interpretability has raised concerns. The inherent black-box nature of AI makes it challenging for human experts to interpret the AI prediction and determine whether AI made the correct decision based on erroneous reasoning [182]. Explainable AI (XAI) techniques aim to demystify the underlying logic and make the AI algorithms more transparent, explainable and trustable [182,183]. Many XAI approaches, such as gradient-weighted class activation mapping (Grad-CAM) and DeConvNet, have been proposed. These methods can reveal the features that contribute to the decision-making process. For example, Grad-CAM and DeConvNet can generate heatmaps to highlight the contributing regions of the input images [184,185,186]. Hopefully, these methods can be more extensively applied to orthodontics-related AI models in the future [182].

Last but not least, overfitting is a common issue in the whole field of AI. This means the model performs excessively well in the training datasets but shows unsatisfactory performance in the testing dataset [187]. Factors like data insufficiency, low data heterogeneity and excessive variables could all lead to overfitting [188]. Methods like improving data samples, data augmentation, regularization, cross-validation and specific algorithms have all been reported to prevent overfitting [171,189,190,191]. However, not all the studies reported in this review have taken measurements to avoid overfitting.

Although AI has been extensively explored in orthodontic treatment, there are still several other areas where it could be further investigated, for example, the automated detection of orthodontic treatment needs like the index of orthodontic treatment need (IOTN) and index of orthognathic functional treatment need (IOFTN) [192,193]. Currently, AI excels mostly in orthodontics diagnosis, yet it has limited guidance on the treatment process. Orthodontists may encounter various challenges throughout the entire orthodontic treatment, including correcting deep overbites and avoiding bone dehiscence or fenestration. Using AI to aid in preventing or addressing these issues could also be a potential area for future development. As clinical data continue to grow and AI computing power improves, there is no doubt that AI will significantly advance the field of orthodontics.

## 4. Conclusions

AI has shown manifold applications in orthodontics, contributing to diagnosis, treatment planning and clinical practice. At present, AI still cannot fully replace human experts, but it can serve as a quality-assuring component in clinical routine. With improvement in data availability, computing power and analytics methods, it is believed that AI can better assist clinical orthodontic care.

## Figures and Tables

**Table 1 healthcare-11-02760-t001:** The application of AI in cephalometric analysis in the past 5 years.

Author (Year)	Data Type	Dataset Size (Training/Test)	No. of Landmarks/Measurements	Algorithm	Performance
Payer et al. (2019) [28]	Lateral cephalograms	150/250	19/0	CNN	Error radii: 26.67% (2 mm), 21.24% (2.5 mm), 16.76% (3 mm), and 10.25% (4 mm).
Nishimoto et al. (2019) [29]	Lateral cephalograms	153/66	10/12	CNN	Average prediction errors: 17.02 pixels. Median prediction errors: 16.22 pixels.
Zhong et al. (2019) [30]	Lateral cephalograms	150/100 (additional 150 images than validation set).	19/0	U-Net	Test 1: MRE: 1.12 ± 0.88 mm. SDR within 2, 2.5, 3, and 4 mm: 86.91%, 91.82%, 94.88%, and 97.90%, respectively. Test 2: MRE: 1.42 ± 0.84 mm. SDR within 2, 2.5, 3, and 4 mm: 76.00%, 82.90%, 88.74%, and 94.32%, respectively.
Park et al. (2019) [31]	Lateral cephalograms	1028/283	80/0	YOLOv3, SSD	YOLOv3 demonstrated overall superiority over SSD in terms of accuracy and computational performance. For YOLOv3, SDR within 2, 2.5, 3, and 4 mm: 80.40%, 87.4%, 92.00%, and 96.2%, respectively.
Moon et al. (2020) [32]	Lateral cephalograms	Training: 50, 100, 200, 400, 800, 1200, 1600, 2000. Test: 200.	19, 40, 80	CNN (YOLOv3)	The accuracy of AI is positively correlated with the number of training datasets and negatively correlated with the number of detection targets.
Hwang et al. (2020) [33]	Lateral cephalograms	1028/283	A total of 80	CNN (YOLOv3)	Mean detection error: 1.46 ± 2.97 mm.
Oh et al. (2020) [34]	Lateral cephalograms	150/100 (additional 150 images than validation set).	19/8	CNN (DACFL)	MRE: 14.55 ± 8.22 pixel. SDR within 2, 2.5, 3, and 4 mm: 75.9%, 83.4%, 89.3%, and 94.7%, respectively. Classification accuracy: 83.94%.
Kim et al. (2020) [35]	Lateral cephalograms	1675/400	23/8	Stacked hourglass deep learning model.	Point-to-point error: 1.37 ± 1.79 mm. SCR: 88.43%.
Kunz et al. (2020) [36]	Lateral cephalograms	1792/50	18/12	CNN	The CNN models showed almost no statistically significant differences with the humans’ gold standard.
Alqahtani et al. (2020) [37]	Lateral cephalograms	-/30	16/16	Commercially available web-based platform (CephX, https://www.orca-ai.com/, accessed on 23 August 2023)	The results obtained from CephX and manual landmarking did not exhibit clinically significant differences.
Lee et al. (2020) [38]	Lateral cephalograms	150/250	19/8	Bayesian CNN	Mean landmark error: 1.53 ± 1.74 mm. SDR within 2, 3, and 4 mm: 82.11%, 92.28%, and 95.95%, respectively. Classification accuracy: 72.69~84.74.
Yu et al. (2020) [39]	Lateral cephalograms	A total of 5890	Four skeletal classification indicators.	Multimodal CNN	Sensitivity, specificity, and accuracy for vertical and sagittal skeletal classification: >90%.
Li et al. (2020) [40]	Lateral cephalograms	150/100 (additional 150 images than validation set).	19/0	GCN	MRE: 1.43 mm. SDR within 2, 2.5, 3, and 4 mm: 76.57%, 83.68%, 88.21%, and 94.31%, respectively.
Tanikawa et al. (2021) [41]	Lateral cephalograms	1755/30 for each subgroup	26/0	CNN	Mean success rate: 85~91%. Mean identification error: 1.32~1.50 mm.
Zeng et al. (2021) [42]	Lateral cephalograms	150/100 (additional 150 images than validation set).	19/8	CNN	MRE: 1.64 ± 0.91 mm. SDR within 2, 2.5, 3, and 4 mm: 70.58%, 79.53%, 86.05%, and 93.32%, respectively. SCR: 79.27%.
Kim et al. (2021) [24]	Lateral cephalograms	2610/100 (additional 440 images than validation set)	20/0	Cascade CNN	Overall detection error: 1.36 ± 0.98 mm.
Hwang et al. (2021) [43]	Lateral cephalograms	1983/200	19/8	CNN (YOLOv3)	SDR within 2, 2.5, 3, and 4 mm: 75.45%, 83.66%, 88.92%, and 94.24%, respectively. SCR: 81.53%.
Bulatova et al. (2021) [44]	Lateral cephalograms	-/110	16/0	CNN (YOLOv3) (Ceppro software)	Total of 12 out of 16 points showed no statistical difference in absolute differences between AI and manual landmarking.
Jeon et al. (2021) [45]	Lateral cephalograms	-/35	16/26	CNN	None of the measurements showed statistically differences except the saddle angle, linear measurements of maxillary incisor to NA line and mandibular incisor to NB line.
Hong et al. (2022) [46]	Lateral cephalograms	3004/184	20/	Cascade CNN	Total mean error was 1.17 mm. Accuracy percentage: 74.2%.
Le et al. (2022) [47]	Lateral cephalograms	1193/100	41/8	CNN (DACFL)	MRE of 1.87 ± 2.04 mm. SDR within 2, 2.5, 3, and 4 mm: 73.32%, 80.39%, 85.61%, and 91.68%, respectively. Average SCR: 83.75%.
Mahto et al. (2022) [48]	Lateral cephalograms	-/30	18/12	Commercially available web-based platform (WebCeph, https://webceph.com, accessed on 23 August 2023)	Intraclass correlation coefficient: 7 parameters >0.9 (excellent agreement), 5 parameters: 0.75~0.9 (good agreement).
Uğurlu et al. (2022) [49]	Lateral cephalograms	1360/180 (additional 140 images than validation set)	21/0	CNN (FARNet)	MRE: 3.4 ± 1.57 mm. SDR within 2, 2.5, 3, 4 mm: 76.2%, 83.5%, 88.2%, 93.4%, respectively.
Yao et al. (2022) [50]	Lateral cephalograms	312/100 (additional 100 images than validation set)	37/0	CNN	MRE: 1.038 ± 0.893 mm. SDR within 1, 1.5, 2, 2.5, 3, 3.5, 4 mm: 54.05%, 91.89%, 97.30%, 100%, 100%, 100%, respectively.
Lu et al. (2022) [51]	Lateral cephalograms	150/250	19/0	GCN	MRE: 1.19 mm. SDR within 2, 2.5, 3, and 4 mm: 83.20%, 88.93%, 92.88%, and 97.07%, respectively.
Tsolakis et al. (2022) [52]	Lateral cephalograms	-/100	16/18	CNN (commercially available software: CS imaging V8).	Differences between the AI software (CS imaging V8) and manual landmarking were not clinically significant.
Duran et al. (2023) [53]	Lateral cephalograms	-/50	32/18	Commercially available web-based platform (OrthoDx, https://ortho dx.phime ntum.com; WebCeph, https://webceph.com, accessed on 23 August 2023)	Consistency between AI software and manual landmarking: A statistically significant good level: angular measurements; a weak level: linear measurement and soft tissue parameters.
Ye et al. (2023) [54]	Lateral cephalograms	-/43	32/0	Commercially available software (MyOrthoX, Angelalign, and Digident)	MRE: MyOrthoX: 0.97 ± 0.51 mm. Angelalign: 0.80 ± 0.26 mm. Digident: 1.11 ± 0.48 mm. SDR (%) (within 1/1.5/2 mm): MyOrthoX: 67.02 ± 10.23/82.80 ± 7.36/89.99 ± 5.17. Angelalign: 78.08 ± 14.23/89.29 ± 14.02/93.09 ± 13.64. Digident: 59.13 ± 10.36/78.72 ± 5.97/87.53 ± 4.84.
Ueda et al. (2023) [55]	Lateral cephalometric data	A total of 220	0/8	RF	Overall accuracy: 0.823 ± 0.060.
Bao et al.(2023) [56]	Reconstructed lateral cephalograms from CBCT	-/85	19/23	Commercially available software (Planmeca Romexis 6.2)	For landmarks: MRE: 2.07 ± 1.35 mm SDR within 1, 2, 2.5, 3, and 4 mm: 18.82%, 58.58%, 71.70%, 82.04%, and 91.39%, respectively. For measurements: The rates of consistency within the 95% limits of agreement: 91.76~98.82%.
Kim et al. (2021) [57]	Reconstructed Posteroanterior cephalograms from CBCT	345/85	23/0	Multi-stage CNN	MRE: 2.23 ± 2.02 mm SDR within 2 mm: 60.88%.
Takeda et al. (2021) [58]	Posteroanterior cephalograms	320/80	4/1	CNN, RF	The CNN showed higher coefficient of determination than RF and less mean absolute error for the distance from the vertical reference line to menton. CNN with a stochastic gradient descent optimizer had the best performance.
Lee et al. (2019) [59]	CBCT	20/7	7	Deep learning	Average point-to-point error: 1.5 mm.
Torosdagli et al. (2019) [60]	CBCT	A total of 50	9/0	Deep geodesic learning	Errors in the pixel space: <3 pixels for all landmarks.
Yun et al. (2020) [61]	CBCT	230/25	93/0	CNN	Average point-to-point error: 3.63 mm.
Kang et al. (2021) [62]	CT	20/8	16/0	Multi-stage DRL	Mean detection error: 1.96 ± 0.78. SDR within 2, 2.5, 3, and 4 mm: 58.99%, 75.39%, 86.52%, and 95.70%, respectively.
Ghowsi et al. (2022) [63]	CBCT	-/100	53/0	Commercially available software (Stratovan Corporation)	Mean absolute error: 1.57 mm. Mean error distance: 3.19 ± 2.6 mm. SDR within 2, 2.5, 3, and 4 mm: 35%, 48%, 59%, and 75%, respectively.
Dot et al. (2022) [64]	CT	128/38 (additional 32 images as validation set).	33/15	SCN	For landmarks: MRE: 1.0 ± 1.3 mm. SDR within 2, 2.5, and 3 mm: 90.4%, 93.6%, and 95.4%, respectively. For measurements: Mean errors: −0.3 ± 1.3° (angular), −0.1 ± 0.7 mm (linear).
Blum et al. (2023) [65]	CBCT	931/114	35/0	CNN	Mean error: 2.73 mm.

MRE, mean radial error; SDR, success detection rate; YOLOv3, You-Only-Look-Once version 3; SSD, Single-Shot Multibox Detector; SCR, success classification rates; DACFL, deep anatomical context feature learning; CBCT, cone-beam computed tomography; GCN, graph convolutional networks, FARNet, feature aggregation and refinement network; DRL, deep reinforcement learning; CT, computerized tomography; SCN, SpatialConfiguration-Net.

**Table 2 healthcare-11-02760-t002:** The application of AI in skeletal maturation assessment using lateral cephalograms.

Author (Year)	Data Type	Dataset Size (Training/Test)	Algorithm	Performance
Kök et al. (2019) [99]	Lateral cephalograms	240/60	k-NN, NB, DT, ANN, SVM, RF, LR	Mean rank of accuracy: k-NN: 4.67, NB: 4.50, DT: 3.67, ANN: 2.17, SVM: 2.50, RF: 4.33, LR: 5.83.
Makaremi et al. (2019) [100]	Lateral cephalograms	Training: 360/600/900/1870 Evaluation: 300 Testing: 300	CNN	Performance varied depending on image numbers and pre-processing method.
Amasya et al. (2020) [101]	Lateral cephalograms	498/149	LR, SVM, RF, ANN, DT	Accuracy: LR: 78.69%, SVM: 81.08%, RF: 82.38%, ANN: 86.93%, DT: 85.89%.
Amasya et al. (2020) [102]	Lateral cephalograms	-/72	ANN	Average of 58.3% agreement with four human observers.
Kök et al. (2021) [91]	Lateral cephalograms	A total of 419	Total of 24 different ANN models	The highest accuracy was 0.9427.
Seo et al. (2021) [103]	Lateral cephalograms	A total of 600	ResNet-18, MobileNet-v2, ResNet-50, ResNet-101, Inception-v3, Inception-ResNet-v2	Accuracy/Precision/Recall/F1 score: ResNet-18: 0.927 ± 0.025/0.808 ± 0.094/0.808 ± 0.065/0.807 ± 0.074. MobileNet-v2: 0.912 ± 0.022/0.775 ± 0.111/0.773 ± 0.040/0.772 ± 0.070. ResNet-50: 0.927 ± 0.025/0.807 ± 0.096/0.808 ± 0.068/0.806 ± 0.075. ResNet-101: 0.934 ± 0.020/0.823 ± 0.113/0.837 ± 0.096/0.822 ± 0.054. Inception-v3: 0.933 ± 0.027/0.822 ± 0.119/0.833 ± 0.100/0.821 ± 0.082. Inception-ResNet-v2: 0.941 ± 0.018/0.840 ± 0.064/0.843 ± 0.061/0.840 ± 0.051.
Zhou et al. (2021) [104]	Lateral cephalograms	980/100	CNN	Mean labeling error: 0.36 ± 0.09 mm. Accuracy: 71%.
Kim et al. (2021) [105]	Lateral cephalograms	480/120	CNN	Three-step model obtained the highest accuracy at 62.5%.
Rahimi et al. (2022) [106]	Lateral cephalograms	692/99 (additional 99 images than validation set).	ResNet-18, ResNet-50, ResNet-101, ResNet-152, VGG19, DenseNet, ResNeXt-50, ResNeXt-101, MobileNetV2, InceptionV3.	ResNeXt-101 showed the best test accuracy: Six-class: 61.62%, three-class: 82.83%.
Radwan et al. (2023) [107]	lateral cephalograms	1201/150 (additional 150 images than validation set).	U-Net, Alex-Net	Segmentation network: Global accuracy: 0.99. Average dice score: 0.93. Classification network: Accuracy: 0.802. Sensitivity (pre-pubertal/pubertal/post-pubertal): 0.78/0.45/0.98. Specificity (pre-pubertal/pubertal/post-pubertal): 0.94/0.94/0.75. F1 score (pre-pubertal/pubertal/post-pubertal): 0.76/0.57/0.90.
Akay et al. (2023) [98]	lateral cephalograms	352/141 (additional 94 images than validation set).	CNN	Classification accuracy: 58.66%. Precision (stage 1/2/3/4/5/6): 0.82/0.47/0.64/0.52/0.55/0.52. Recall (stage 1/2/3/4/5/6): 0.70/0.74/0.58/0.54/0.37/0.60. F1 score (stage 1/2/3/4/5/6): 0.76/0.57/0.61/0.53/0.44/0.56.

k-NN, k-nearest neighbors; NB, Naive Bayes; LR, logistic regression; CNN, convolutional neural network; SVM, support vector machine; RF, random forest; ANN, artificial neural network; DT, decision tree.

**Table 3 healthcare-11-02760-t003:** The application of AI in upper-airway obstruction assessment.

Author (Year)	Data Type	Dataset Size (Training/Test)	Algorithm	Purpose	Performance
Shen et al. (2020) [109]	Lateral cephalograms	488/116 (additional 64 images than validation set)	CNN	Adenoid hypertrophy detection	Classification accuracy: 95.6%. Average AN ratio error: 0.026. Macro F1 score: 0.957.
Zhao et al. (2021) [110]	Lateral cephalograms	581/160	CNN	Adenoid hypertrophy detection	Accuracy: 0.919. Sensitivity: 0.906. Specificity: 0.938. ROC: 0.987.
Liu et al. (2021) [111]	Lateral cephalograms	923/100	VGG-Lite	Adenoid hypertrophy detection	Sensitivity: 0.898. Specificity: 0.882. Positive predictive value: 0.880. Negative predictive value: 0.900. F1 score: 0.889.
Sin et al. (2021) [112]	CBCT	214/46 (additional 46 images than validation set)	CNN	Pharyngeal airway segmentation	Dice ratio: 0.919. Weighted IoU: 0.993.
Leonardi et al. (2021) [113]	CBCT	20/20	CNN	Sinonasal cavity and pharyngeal airway segmentation	Mean matching percentage (tolerance 0.5 mm/1.0 mm): 85.35 ± 2.59/93.44 ± 2.54
Shujaat et al. (2021) [114]	CBCT	48/25 (additional 30 images than validation set)	3D U-Net	Pharyngeal airway segmentation	Accuracy: 100%. Dice score:0.97 ± 0.02. IoU: 0.93 ± 0.03.
Jeong et al. (2023) [115]	Lateral cephalograms	1099/120	CNN	Upper-airway obstruction evaluation	Sensitivity: 0.86. Specificity: 0.89. Positive predictive value: 0.90. Negative predictive value: 0.85, F1 score: 0.88.
Dong et al. (2023) [116]	CBCT	A total of 87	HMSAU-Net and 3D-ResNet	Upper-airway segmentation and adenoid hypertrophy detection	Segmentation: Dice value: 0.96. Diagnosis: accuracy: 0.912. Sensitivity: 0.976. Specificity: 0.867. Positive predictive value: 0.837. Negative predictive value: 0.981. F1 score: 0.901.
Jin et al. (2023) [117]	CBCT	A total of 50	Transformer and U-Net	Nasal and pharyngeal airway segmentation	Precision: 85.88~94.25%. Recall: 93.74~98.44%. Dice similarity coefficient: 90.95~96.29%. IoU: 83.68~92.85%.

ROC, receiver operating characteristic; CBCT, cone-beam computed tomography; CNN, convolutional Neural Network; AN, adenoid–nasopharynx; IoU, Intersection over Union; HMSAU-Net, hierarchical masks self-attention U-net; 3D, three-dimensional.

**Table 4 healthcare-11-02760-t004:** The application of AI in treatment planning.

Author (Year)	Data Type	Dataset Size (Training/Test)	Algorithms	Purpose	Performance
Xie et al. (2010) [121]	Cephalometric variables, cast measurement.	180/20	ANN	To predict tooth extraction diagnosis.	Accuracy: 80%.
Jung et al. (2016) [122]	Cephalometric variables, dental variable, profile variables, and chief complaint for protrusion.	64/60 (additional 32 samples than validation set)	ANN	To predict tooth extraction diagnosis, and extraction patterns.	Success rate: Tooth extraction diagnosis: 93%. Extraction patterns: 84%.
Li et al. (2019) [123]	Demographic data, cephalometric data, dental data, and soft tissue data.	A total of 302 samples	MLP (ANN)	To predict tooth extraction diagnosis, extraction patterns and anchorage patterns.	Accuracy: For extraction diagnosis: 94%. For extraction patterns: 84.2%. For anchorage patterns: 92.8%.
Suhail et al. (2020) [124]	Diagnosis, feature identification of photos, models and X-rays.	A total of 287 samples	ANN, LR, RF	To predict tooth extraction diagnosis, and extraction patterns.	For extraction diagnosis: LR outperformed the ANN. For extraction patterns: RF outperformed ANN.
Etemad et al. (2021) [125]	Demographic data, cephalometric data, dental data, and soft tissue data.	A total of 838 samples	RF, MLP (ANN)	To predict tooth extraction diagnosis.	Accuracy of RF with 22/117/all inputs: 0.75/0.76/0.75. Accuracy of MLP with 22/117/all inputs: 0.79/0.75/0.79.
Shojaei et al. (2022) [126]	Medical records, extra and intra oral photos, dental model records, and radiographic images.	A total of 126 samples	LR, SVM, DT, RF, Gaussian NB, KNN Classifier, ANN	To predict tooth extraction diagnosis, extraction patterns and anchorage patterns.	Accuracy for extraction decision: ANN: 93%, LR:86%, SVM:83%, DT: 76%, RF: 83%, Gaussian NB: 72%, KNN Classifier: 72%. Accuracy for extraction pattern: ANN: 89%, RF:40%. Accuracy for extraction and anchorage pattern: ANN: 81%, RF:23%.
Real et al. (2022) [127]	Sex, model variables, cephalometric variables, outcome variables.	-/214	Commercially available software (Auto-WEKA)	To predict tooth extraction diagnosis.	Accuracy: 93.9%: input model and cephalometric data. 87.4%: input only model data. 72.7%: input only cephalometric data.
Leavitt et al. (2023) [128]	Cephalometric variables, dental variables, demographic characteristics.	256/110	RF, LR, SVM	To predict tooth extraction patterns.	Overall accuracy: RF: 54.55%, SVM: 52.73%, LR: 49.09%.
Ryu et al. (2023) [78]	Intraoral photographs, extraction decision.	2736/400	ResNet (ResNet50, ResNet101), VggNet (VGG16, and VGG19)	To predict tooth extraction diagnosis.	Accuracy: Maxilla: VGG19 (0.922) > ResNet101 (0.915) > VGG16 (0.910) > ResNet50 (0.909). Mandible: VGG19 (0.898) = VGG16 (0.898) > ResNet50 (0.895) > ResNet101 (0.890).
Prasad et al. (2022) [129]	Clinical data, cephalometric data, cast and photographic data.	A total of 700 samples	RF, XGB, LR, DT, K-Neighbors, Linear SVM, NB	To predict skeletal jaw base, extraction diagnosis for Class 1 jaw base, and functional/camouflage/surgical strategies for Class 2/3 jaw base.	Different algorithms showed different accuracies in different layers. RF performed best in 3 out of 4 layers.
Knoops et al. (2019) [130]	3D face scans	A total of 4261	SVM for classification LR, RR, LARS, and LASSO for regression	To predict surgery/non-surgery decision and surgical outcomes.	For surgery/non-surgery decision: Accuracy: 95.4%. Sensitivity: 95.5%. Specificity: 95.2%. For surgical outcomes simulation: Average error: LARS and RR (1.1 ± 0.3 mm). LASSO (1.3 ± 0.3 mm). LR (3.0 ± 1.2 mm).
Choi et al. (2019) [131]	Lateral cephalometric variables, dental variable, profile variables, chief complaint for protrusion.	136/112 (additional 68 samples than validation set)	ANN	To predict surgery/non-surgery decision, extraction/non-extraction for surgical treatment.	Accuracy for all dataset: Diagnosis of surgery/non-surgery: 96%. Diagnosis of extraction/non-extraction for Class II surgery: 97%. Diagnosis of extraction/non-extraction for Class III surgery: 88%. Diagnosis of extraction/non-extraction for surgery: 91%.
Lee et al. (2020) [132]	Lateral cephalograms.	220/40 (additional 73 samples than validation set)	CNN (Modified-Alexnet, MobileNet, and Resnet50)	To predict the need for orthognathic surgery.	Average accuracy for all dataset: Modified-Alexnet: 96.4%. MobileNet: 95.4%. Resnet50: 95.6%.
Jeong et al. (2020) [133]	Facial photos (front and right).	A total of 822 samples. Group 1: 207/204. Group 2: 205/206	CNN	To predict the need for orthognathic surgery.	Accuracy: 0.893. Precision: 0.912. Recall: 0.867. F1 scores:0.889.
Shin et al. (2021) [134]	Lateral cephalograms and posteroanterior cephalograms.	A total of 840 samples. Group 1: 273/304 (additional 30 samples than validation set). Group 2: 98/109 (additional 11 samples than validation set)	CNN	To predict the diagnosis of orthognathic surgery.	Accuracy: 0.954. Sensitivity: 0.844. Specificity: 0.993.
Kim et al. (2021) [135]	Lateral cephalograms.	810/150	CNN (ResNet-18, 34, 50, 101)	To predict the diagnosis of orthognathic surgery.	Accuracy for test dataset: ResNet-18/34/50/101: 93.80%/93.60%/91.13%/91.33%.
Lee et al. (2022) [136]	Cephalometric measurements, demographic characteristics, dental analysis, and chief complaint.	136/60	RF, LR	To predict the diagnosis of orthognathic surgery.	Accuracy (RF/LR): 90%/78%. Sensitivity (RF/LR): 84%/89%. Specificity (RF/LR): 93%/73%.
Woo et al. (2023) [137]	Intraoral scan data	-/30	Three commercially available software packages (Autolign, Outcome Simulator Pro, Ortho Simulation)	To evaluate the accuracy of automated digital setup accuracy.	Mean error of three pieces of software: Linear movement: 0.39~1.40 mm. Angular movement: 3.25~7.80°.
Park et al. (2021) [138]	Lateral cephalograms	A total of 284 cases	CNN (U-Net)	To predict the cephalometric changes of Class II patients after using modified C-palatal plates.	Total mean error: 1.79 ± 1.77 mm.
Tanikawa et al. (2021) [139]	3D facial images	A total of 72 cases in surgery group and 65 cases in extraction group	Deep learning	To predict facial morphology change after orthodontic or orthognathic surgical treatment.	Average system errors: Surgery group: 0.94 ± 0.43 mm; orthodontic group: 0.69 ± 0.28 mm. Success rates (<1 mm): Surgery group: 54%; orthodontic group: 98%. Success rates (<2 mm): Surgery group: 100%; orthodontic group: 100%.
Park et al. (2022) [140]	CBCT	268/44	cGAN	To predict post-orthodontic facial changes.	Mean prediction error: 1.2 ± 1.01 mm. Accuracy within 2 mm: 80.8%.
Xu et al. (2022) [141]	Total of 17 clinical features	A total of 196 cases	ANN	To predict patient experience of Invisalign treatment.	Predictive success rate: Pain: 87.7%. Anxiety: 93.4%. Quality of life: 92.4%.

ANN, artificial neural network; DT, decision tree; RF, random forest; LR, logistic regression; SVM, support vector machine; NB, naive bayes; KNN, k-nearest neighbors; MLP, multilayer perceptron; XGB, eXtreme Gradient Boosting; RR, ridge regression; LARS, least-angle regression; LASSO, least absolute shrinkage and selection operator regression; CNN, convolutional neural network; CBCT, cone-beam computed tomography; cGAN, conditional generative adversarial networks.

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
