# Peer review of "Application of Artificial Intelligence in Orthodontics: Current State and Future Perspectives"

_healthcare, 2023, doi:10.3390/healthcare11202760_

Round 1
Reviewer 1 Report
Dear colleagues,
congratulations of this very interesting manuscript.
Please consider these comments for revision:
1. ll. 23/24: This sentence is not really comprehensible. Maybe formulate it as: “…, the field of AI has a lot of potential as AI can be employed to solve a variety of tasks.”
2. ll. 36-45: Please describe the most prominent differences between ANN and CNN and why CNN are usually preferred for computer vision and image-dependant machine learning.
3. Ll. 51/52 This bold statement needs to be underlined by sufficient source.
4. Ll. 54/55 Please add a source here.
5. Ll. 57/58 Please add a source here.
6. Ll. 65 Please add source when you mention other reviews.
7. Please add a section Material and Methods to describe your search strategy, inclusion and exclusion criteria etc. See: https://www.equator-network.org/reporting-guidelines/prisma/
8. Ll. 76-80 Please add a source here.
9. Ll. 84-86 Please add a source here.
10. Ll. 88-90 Please add a source here.
11. Ll. 92-83 Please explain in M&M how you selected the literature and how you assessed the quality of each paper
12. Ll. 100-102 What about homogeneity of the results? Was any statistic test applicable?
13. Ll. 120/121 Please quickly describe why these processes are drawbacks (time consuming, etc.).
14. L. 134 Please explain this term to the reader in one short sentence.
15. L. 141 Maybe also explain the differences in technique between CNN application on radiographs and photos.
16. Ll. 153/154 Are there any publications on these "automated measurements"? How do they work?
17. Ll. 163/164 Please describe you search strategy and criteria. You wrote that you concentrated on recent articles...
18. Ll. 169-171 What are the results of these experiments?
19. Ll. 179/180 Please add a source here.
20. L. 181 Please explain this method.
21. Ll. 190/191 Interesting finding. Is there a hypothesis on why this was the case?
22. L. 290 Please add a short summary for this paragraph.
23. Ll. 212/123 Please add a source and explain why upper airway obstruction is important for facial growth and development (one sentence).
24. Ll. 247/248 Please give a source here.
25. Ll. 255/256 Please give a source here.
26. Ll. 269/270 Please give a little more context and background information on these two algorithms.
27. L. 274 Please explain SVM in a short sentence.
28. L. 282 Please add a short summary for this paragraph.
29. Ll. 285/286 Please give a source here.
30. L. 305 Please add a short summary for this paragraph.
31. Ll. 308/310 Please give a source here.
32. Ll. 313/314 Please give a source here.
33. Ll. 317/319 Please give a source here.
34. Ll. 331/332 Please give a short explanation of cGAN
35. L. 342 Please add a short summary for this paragraph.
36. Ll. 346/347 Please give a source here.
37. L. 357 Please give a short explanation of 3D-UnetSE
38. Ll. 371-373 Please give a source here.
39. L. 397 Please give a short explanation of DeepID
40. Ll. 415/417 Please give a short explanation for all of these
41. Ll. 419/420 Please explain Blockchain technology in a short sentence.
42. Ll. 436-438 Please explain a little
43. Ll. 445/446 Would that not be a good criteria to judge the quality of the presented papers?
I believe, that this review will give a thorough overview of the current state of AI application in orthodontics and will help improve research quality and comparability.
Nonetheless, some revisions are necessary before the final acceptance.
The quality of English is sufficient.
Reviewer 2 Report
see attached

Reviewer 3 Report
Paper needs revision for grammar and. Correction of numerous typos in the text
can you add the major findings to the abstract, you can use subheadings
I suggest you present the paper in systematic approach and define PICO
also you can decide this into 2 papers one based on diagnosis and the other one treatment plan
line 36-50 , definition of malocclusion is not accurate , correct to Malocclusion is a deviation form norm and manifestation of normal biological variability (Eur J Orthod. 2009 31(5):477-84.)
mention that nearly 30% of any population present with great need for orthodontic treatment (World J Orthod. 2010 ;11(1):43-8)
section 2.1.2 dental analysis is not written well, revise
in future prospective suggest programs that detect orthodontic treatment needs like IOTN (Prog Orthod. 2011;12(2):132-42; ) or IOFTN for orthognathic treatment need indices (J Plast Reconstr Aesthet Surg. 2016 Jun;69(6):796-801)
Paper needs revision
Round 2
Reviewer 3 Report
Thank you for the revisions, a good contribution to AI in orthodontics
Minor editing would be beneficial